# Job Dissatisfaction and Its Predictors among Healthcare Workers of ‘Type 2 Health Clinics’ in North-Eastern Malaysia

**DOI:** 10.3390/ijerph192316106

**Published:** 2022-12-01

**Authors:** Mohd Ikhwan Azmi, Aziah Daud, Mohd Nazri Shafei, Anees Abdul Hamid

**Affiliations:** 1Department of Community Medicine, School of Medical Sciences, Health Campus, Universiti Sains Malaysia, Kota Bharu 16150, Kelantan, Malaysia; 2Kelantan State Health Department, Ministry of Health Malaysia, Kota Bharu 15200, Kelantan, Malaysia

**Keywords:** job dissatisfaction, primary health clinic, factor associated, north-eastern Malaysia

## Abstract

It is crucial to comprehend factors associated to job dissatisfaction among healthcare workers (HCWs) in Malaysia’s primary health clinics, especially those working in ‘Type 2 Health Clinics’ which cater for populations of >50,000 and a daily average number of patients between 500 and 800. It is essential to ensure that effective strategies can be proposed to promote job satisfaction. A total of 314 HCWs from ‘Type 2 Health Clinics’ in north-eastern Malaysia consented to participate in this cross-sectional study, conducted between October 2020 and December 2021. The Job Satisfaction Survey was used to assess job dissatisfaction. The prevalence of job dissatisfaction was 35.7%. The significant factors associated with job dissatisfaction were younger age and those who were dissatisfied with their yearly performance mark. Targeted interventional activities for young HCWs and for those who are dissatisfied with their yearly performance mark are recommended to improve job satisfaction.

## 1. Introduction

Job dissatisfaction consists of one’s cognitive, emotional, and behavioral response to their job [1]. It was discovered that agreeableness (the traits of cooperation and likeableness), extraversions (the traits of being assertive, enthusiastic, and energetic), and conscientiousness (the traits of high level of organization, hard work, and goal persuasion) are related to job satisfaction, whereas neuroticism (the traits of lack of emotional stability and lack of positive psychological adjustment) and lack of openness to experience (the trait of unconventionality) are related to job dissatisfaction [2]. Apart from that, job dissatisfaction also has been linked to workers’ level of autonomy in how they act, given their skill set and work expectations. It also has to do with employees’ psychological challenges in carrying out their duties [3]. Job dissatisfaction can have a negative impact on both the organization and the people receiving the services. It can jeopardize patients’ safety and treatment, being related to poor job performance and absenteeism amongst healthcare workers (HCWs) [4]. Hence, it is important to determine job dissatisfaction, especially among HCWs.

In Malaysia, the primary health clinic is an important structure in the public healthcare system as it provides curative, promotive, and rehabilitative care services. However, there is a noticeable shortage of HCWs in primary health clinics, as most of them leave for the private sector or leave the healthcare system entirely [5]. This can lead to an imbalance between those who provide healthcare services and those who require healthcare services, which contributes to occupational stress and leads to job dissatisfaction, which later leads to quitting [6,7]. Apart from that, poor job performance may jeopardize patients’ safety and care [4]. Hence, it is important to determine the prevalence of job dissatisfaction and to identify its predictors among HCWs in Malaysia.

## 2. Literature Review

### 2.1. Job Dissatisfaction and Healthcare Workers

Internationally, two studies conducted in Ethiopia found that the prevalence of job dissatisfaction was around 46% [8,9]. However, Behmann et al. (2012) found that the prevalence of job dissatisfaction among primary care physicians in Germany was slightly lower at 36% [10]. In a local Malaysian context, Manan et al. (2015) reported that 48% of pharmacists in Negeri Sembilan, Selangor, and Perak were unsatisfied with their job, and found that HCWs aged 35 and older and who had worked more than seven years’ experience are more likely to be satisfied with their jobs [11]. Aidalina M. (2015), who studied the brain drain phenomenon of physicians in the public and private sector in Selangor and Kuala Lumpur, found that 35.6% were dissatisfied with their work and 55% of those respondents felt neutral about public-sector job satisfaction, driving them to leave the public sector to work in the private sector [12].

Numerous factors can contribute to job dissatisfaction, such as co-workers, supervisors, work, and promotion [13]. Job dissatisfaction was found to be associated with age when being both a young and an old worker (near retirement age) [11,14]. Interestingly, looking at the years of employment, it was noted that the longer the worker’s work experience, the more satisfied they were with their job [14,15]. However, it was also found that those who work for more than ten years are less satisfied than young workers [10]. Other than that, gender dominance is subjected to conflicting studies in regards to job dissatisfaction. There were studies that found no gender difference with job dissatisfaction [10,11,14] and studies that also found that females were more prone to job dissatisfaction than males [12,16].

One of the elements that influenced job satisfaction for public servants in Malaysia was satisfaction with their yearly performance mark, which is an annual assessment report, often known as ‘Laporan Nilaian Prestasi Tahunan’ (LNPT). LNPT, according to Malaysia’s Public Service Department, aims to improve employee motivation and performance and identify employee potential, and can be used for employee promotion, training, and placement, as well as to effect salary increment. The supervisor evaluated the workers once a year as a means of providing feedback in which they try to identify their subordinates’ areas for improvement, assist them with further training, and help them learn new skills to accomplish their job [17]. A study in Indonesia discovered a high link between perceived supervisor support and job satisfaction, indicating a meaningful association [18]. Aside from reducing worker burnout and increasing job happiness, excellent support also helps employees feel more secure in achieving their work goals [19].

Apart from that, job-related factors also contributed to job dissatisfaction [20]. In depth, they can be divided into intrinsic variables of co-workers, supervision, and work itself, and extrinsic variables of salary and advancement [13]. Working environment plays an important role as an unfavorable working environment contributes to job dissatisfaction [10,16]. Aside from that, those working under appropriate and supportive supervision and colleagues are more pleased with their job compared to those who do not [9,21]. In a local context, job-related factors also play a role in job dissatisfaction as a study involving Malaysia’s family physicians in 2016 found that Malaysian family physicians were dissatisfied with their salary, recognition, and their working conditions [22].

### 2.2. Primary Health Clinic in Malaysia

According to the Family Health Development Unit, Ministry of Health, Malaysia under the Primer Infrastructure Development Sector, a health clinic can be divided into seven types depending on its catchment population and daily average number of patients. It provides services such as the out-patient department, accident and emergency, maternal and child health, dental, rehabilitation, radiography, laboratories, and pharmacy [23]. Table 1 shows the types of primary health clinic in Malaysia

## 3. Materials and Methods

### 3.1. Study Design and Population

A cross-sectional study was conducted between October 2020 and December 2021. It was held at all ‘Type 2 Health Clinics’ in north-eastern Malaysia which also have two different clinic working systems. Two of the clinics adopt a shift system, namely Health Clinic A and Health Clinic B, and two adopt a non-shift system, namely Health Clinic C and Health Clinic D. These clinics were chosen as their catchment population was >50,000 and their daily average number of patients was between 500 and 800 [23]. Using a two-proportion formula, with a 95% confidence interval, 5% precision, and allowing a 10% non-response rate or data entry error, the required sample size was 335 (*p* = 0.63) [24]. All available HCWs were included in the study as the required sample size exceeded the number of the workers in those clinics.

### 3.2. Data Collection and Research Tool

A proforma as attached in Appendix A was designed to gather sociodemographic data such as age, education level, and years of employment. The Job Satisfaction Survey (JSS) was used to assess job satisfaction among the respondents. The JSS was developed by Paul E. Spector in 1985 and is commonly used in many fields, including healthcare services. It has nine facets and a 36-item scale to evaluate employee attitudes toward their job and its various aspects. The facets were pay, promotion, contingent rewards, operating procedures, supervision, nature of work, fringe benefits, communication, and co-workers. Each item had six options, varying from “strongly disagree” to “strongly agree” [25]. In this study, the JSS scored the Likert scale of minimum and maximum score as satisfied (144–216) and dissatisfied (36–143). Based on a systemic review conducted in 2003, it was found that the JSS has adequate validity and reliability. The discriminant validity was 0.19–0.59 and convergent validity was 0.61–0.80, whereas the internal consistency was 0.91 and test–retest was 0.71 [26]. The Malay language version was translated by Tan Soo Luan in 2010 with reported similar internal consistency of Cronbach’s α 0.84 when compared to its English version [27]. Permission to use the questionnaire was obtained before conducting the study.

In this study, a HCW was defined as a person whose work involves contact with a patient or with blood or other body fluids from a patient in a healthcare, laboratory, or public safety situation at a health clinic [28]. We included all ‘Type 2 Health Clinics’ HCWs aged 18 and above who had worked for at least three months in the current clinic’s working system. Those who did not complete at least 80% of the questionnaire data were excluded.

### 3.3. Procedure 

The questionnaire was distributed to all HCWs working in all ‘Type 2 Health Clinics’ in north-eastern Malaysia. Initially, the selected health clinics were informed through a letter that was sent through the ‘Kelantan State Health Department’ explaining the study that will be conducted to the clinic’s manager, who will convey the information to all HCWs at their clinic. The research team then scheduled a face-to-face session with the help of the clinic’s manager with their respective HCWs. The session explained the research objectives and procedures, voluntariness, and anonymity. They were then given enough time to consider their participation, signed the consent form, and answered the questionnaire given. Any questions that arose were entertained by the research team during the session.

### 3.4. Data Analyses

For data entry and analysis, IBM SPSS version 26.0 was used. Data were checked and cleaned once they were entered. In descriptive analyses, data were analyzed and presented as frequency (*n*) and percentage (%). Simple and multiple logistic regression was used to find the predictor of job dissatisfaction as the outcome was categorical and binary of either job dissatisfaction or job satisfaction. Forward LR and Backward LR were used to compare and identify the final model of factors associated with job dissatisfaction. It was then checked for multicollinearity, interaction, and the model’s fitness. The final model was presented as a *p*-value and adjusted OR. A *p*-value of less than 0.05 was set as the level of significance. 

### 3.5. Ethical Consideration

Ethical approval for this study was obtained from the Medical Research and Ethics Committee (MREC), Ministry of Health, Malaysia, with identification number NMRR-20-2574-57270 (IIR). Ethical approval was also obtained from the Human Research Ethics Committee (JEPeM), Universiti Sains Malaysia (USM), (USM/JEPeM/2 0110577). Written consent was obtained from participants prior to the study. Only the researcher can access the data and participant anonymity was applied to ensure the confidentiality of the data. 

## 4. Results

The respond rate was 84.9% (314/370). The mean (SD) age of the HCWs was 40.6 (7.81) years old, and duration of employment was 15.7 (7.58) years. A total of 219 (69.7%) of them completed tertiary education and 284 (90.4%) were satisfied with their yearly performance mark. Table 2 shows the sociodemographic characteristics of the respondents.

A total of 69 (22.0%) of the HCWs in the shift clinics’ working system and 43 (13.7%) in the non-shift clinics’ working system in ‘Type 2 Health Clinics’ in north-eastern Malaysia were dissatisfied with their job. Table 3 provides the details.

The top two job dissatisfaction factors among HCWs in ‘Type 2 Health Clinics’ in north-eastern Malaysia according to JSS facets were related to operating conditions (61.8%) and benefits (55.4%). Table 4 provides the details.

The variables from simple logistic regression with a *p*-value less than 0.25, which were age, gender, race, monthly income, and yearly performance mark, were selected and further analyzed using multiple logistic regression to determine the associated factors. Multiple logistic regression shows that age in years (Adj. OR 0.91; 95% CI: 0.83,0.99, *p* = 0.037) and dissatisfaction with yearly performance mark (Adj. OR 14.80; 95% CI: 3.43,63.763, *p* < 0.001) predicts job dissatisfaction. It can be interpretated as the HCWs working in ‘Type 2 Health Clinics’ in north-eastern Malaysia had 9.4% lower odds of job dissatisfaction as they got older, and those dissatisfied with their yearly performance mark were 14.8 times more likely to develop job dissatisfaction after being adjusted for age. Table 5 shows the details for both simple and multiple logistic regressions.

## 5. Discussion

Our study showed that 35.7% of HCWs in ‘Type 2 Health Clinics’ in north-eastern Malaysia were dissatisfied. This finding, however, shows a lower prevalence as compared to the international and local studies where the prevalence of job dissatisfaction ranged from 46% to 48% [8,9]. The prevalence of dissatisfaction in this study may be contributed by the differences in the HCWs studied in terms of health system organization, geographical area, and socioeconomic population that the healthcare workers served. This could also be because a particular HCW was hired based on their qualifications for specific job tasks where the qualifications met the job requirements. According to another local study, management must ensure that their employees are placed based on their skills, qualifications, and abilities. If they are not, job dissatisfaction may arise, and stress and burnout may result [11]. 

Apart from that, looking specifically into the clinics’ working systems, it was found that only 22.0% of respondents were dissatisfied with the shift clinics’ working system and 13.7% with the non-shift system health clinics. Although the shift clinics’ working system was newly introduced in June 2020 amidst the COVID-19 pandemic, the HCWs already seem to be adapting themselves well, as showed by the lower proportion of job dissatisfaction comparing to those in the non-shift clinics’ working system [29]. A study discovered that, although there were varieties of factors that can influence the length of an employee’s adaptation process to a new working system, the process itself usually takes approximately one month to a year [30].

Age was one of the factors linked with job dissatisfaction among HCWs. Both being a young and an old professional (nearing retirement age) were linked to job dissatisfaction. Numerous types of research have been conducted on the topic of age and job dissatisfaction. These studies found that HCWs under 35 are more likely than those 35 and older to be dissatisfied with their jobs [11,14]. However, other studies showed that HCWs beyond the age of 40 are more likely to be dissatisfied with their jobs [10,31]. This study found that age is one of the factors that contributed to job dissatisfaction among HCWs in ‘Type 2 Health Clinics’ in north-eastern Malaysia. It was found that the HCWs with older age were 9.4% less likely to develop job dissatisfaction than those with younger age by a year after being adjusted for the yearly performance mark. Similarly, a study conducted in Greece and Ireland discovered that as one gets older, satisfaction improves [21,32]. This finding may also have been influenced by the fact that as their age increases, the HCWs become more accustomed to the job and working conditions and have more authority over it. This finding was in line with a local study that discovered worker satisfaction was related to working conditions, career development, and freedom from supervision [33].

Laura L. Carstensen’s socioemotional selectivity theory stated that, following ageing, a shift in motivation influences cognitive processing more positively. Hence, older people are more likely to experience positive emotions. Ng and Feldman (2010) found that age is linked to job attitudes, and their meta-analysis found that age is positively related to job satisfaction [34]. Our finding could be attributed to Malaysia’s functioning healthcare system, which is built on time-based promotion, especially for doctors, dental officers, and pharmacists [5]. This method ensures that those who enter the workforce are given a job scope or description appropriate for their entry-level position, allowing them to adjust gradually to the job’s demand. The job scope would gradually increase with seniority and the potential to advance, thus decreasing job dissatisfaction.

This study also discovered that there was a link between job dissatisfaction and yearly performance marks. The yearly performance mark was a type of feedback from a supervisor to a supervisee about their work performance, which influenced promotions, additional training, and pay increase [17]. After being adjusted for age, it was discovered that HCWs in the surveyed health clinics who were unsatisfied with their yearly performance mark were 14.8 times more likely to develop job dissatisfaction than those who were satisfied. Several studies have found a strong correlation between supervisor support and job satisfaction. According to studies conducted in Greece and Ethiopia, healthcare workers who received enough support were more satisfied with their jobs than those who did not receive such support [4,21]. Winarto and Chalidyanto (2020) revealed a substantial link between the job satisfaction of employees with solid supervisory support compared to employees who lacked support. Furthermore, competent supervision assists employees by preventing employee burnout and confidently fulfilling work objectives [18].

Looking into job-related factors, working conditions and benefits were the top two factors from JSS facets that scored the highest proportion related to job dissatisfaction. These findings were in line with several job dissatisfaction studies that found most of the job dissatisfaction was related to working conditions and benefits [10,16]. In this essence, when implementing certain policies and procedures, they must be adaptable in accordance with the worker’s workload and working conditions, for example, by ensuring the physical environment and equipment are suitable for the new policy to be implemented. Aside from that, the HCWs in this study may have found that the benefits that they received from their job were insufficient compared to their workload or to those of a worker from another place. Hence, it was critical to assign appropriate personnel to specific jobs based on their qualifications and skills with comparable benefits as other places, for example by ensuring the benefit was comparable between the public and private healthcare system.

Data from the Human Resources for Health Country Profiles 2015 Malaysia showed that females dominate Malaysia’s healthcare system. They discovered that females make up 97% of nurses, 75% of pharmacists, and 60% of doctors [35]. This study found that 73.9% of our respondents were females and no significant association was found between gender and job dissatisfaction. Similarly, several local and overseas studies also found no significant association between gender and job satisfaction [15,36,37]. A workforce dominated by a certain gender seems to provide greater support and understanding of each other’s physiological and psychological requirements. However, looking at our and other findings, it appears that this does not affect satisfaction or dissatisfaction in the workplace. 

Some studies discovered a link between minority races and job dissatisfaction. Hispanic and Black nurses were found to be more likely than White nurses to resign in a study of nurses in the United States. Black nurses were also more dissatisfied with their jobs than White nurses [38]. Female and non-Bumiputera doctors were three times more likely to leave the public health service in Malaysia. This was due to dissatisfaction with several factors including benefits, compensation, promotions, work operations, family obligations, and personal challenges [12]. However, this study found no association between race and job dissatisfaction. The respondents’ geography and religion could explain this. The majority race in Malaysia and most respondents in this study in the north-eastern region were Malay–Muslim Bumiputera. Hence, they share a standard belief system and practices, leading to increased understanding and tolerance in the workplace.

All in all, a well-defined job scope of a certain job category at the clinic must be established and maintained to improve present healthcare services and increase job satisfaction. Younger workers must be trained regularly to help them appreciate and understand their jobs better, increasing their level of satisfaction. Regular meetings between employees and bosses should be held to review and discuss issues as they arise. Furthermore, any adjustments or new implementations of specific programs must be thoroughly addressed so that all employees understand the objective and why the changes are necessary to gain their complete support and reduce dissatisfaction.

## 6. Conclusions

Job dissatisfaction among HCWs at ‘Type 2 Health Clinics’ in north-eastern Malaysia was moderately high. Actions taken to reduce job dissatisfaction must be targeted at young HCWs and those that are dissatisfied with their yearly performance mark to ensure that job satisfaction improves. It is important to ensure they can retain workers in the healthcare system amidst the shortage of HCWs in public primary healthcare in Malaysia. 

There were a few limitations for this study. First, this study was conducted at a selected type of health clinic according to their catchment population and services provided in north-eastern Malaysia. This can limit the findings on the associated factors since the findings are specific to types of health clinic and location, which may influence the factors. Additionally, it employed primary data collection methods which were influenced by the honesty and accuracy of the respondents’ answers.

Future research can be conducted among healthcare workers at health clinics, particularly those with different working systems such as shift and non-shift working systems, to fully understand the level of dissatisfaction, associated factors, and how to improve it. This type of study can be conducted with a bigger sample size and population to fully comprehend the dissatisfaction among healthcare workers at health clinics. We also recommend future qualitative research to fully grasp why job dissatisfaction occurs and the healthcare workers’ opinions on improving the situation. This permits fresh insights for policymakers to improve job satisfaction.

## Figures and Tables

**Table 1 ijerph-19-16106-t001:** Primary health clinic types in Malaysia.

Health Clinic Type	Catchment Population	Average Number of Patients/Day	Service Provided
Type 1 Health Clinic	>50,000	>800	Out-patient department, accident and emergency, maternal and child health, dental, rehab, X-ray, laboratory, and pharmacy
Type 2 Health Clinic	>50,000	500–800	
Type 3 Health Clinic	>30,000–50,000	300–500	
Type 4 Health Clinic	>20,000–30,000	150–300	Out-patient department, accident and emergency, maternal and child health, dental, rehab, X-ray, laboratory, pharmacy, ABC (alternative birth center) (optional), sick bay (optional)
Type 5 Health Clinic	>10,000–20,000	100–150	Out-patient department, accident and emergency, maternal and child health, dental, mini laboratory, pharmacy, ABC (alternative birth center) (optional), sick bay (optional)
Type 6 Health Clinic	>5000–10,000	50–100	Out-patient department, accident and emergency, maternal and child health, mini laboratory, pharmacy, ABC (alternative birth center) (optional), sick bay (optional)
Type 7 Health Clinic	<5000	<50	

**Table 2 ijerph-19-16106-t002:** Sociodemographic characteristics of the healthcare workers in ‘Type 2 Health Clinics’ in north-eastern Malaysia (*n* = 314).

Variables	*n* (%)	Mean (SD)
Age (year)		40.6 (7.81)
Gender		
Male	82 (26.1)	
Female	232 (73.9)	
Race		
Malay	309 (98.4)	
Non-Malay	5 (1.6)	
Marital status		
Single/divorced	34 (10.8)	
Married	280 (89.2)	
Education level		
Tertiary	219 (69.7)	
Secondary	95 (30.3)	
Health clinic’s working system		
Non-shift	142 (45.2)	
Shift	172 (54.8)	
Number of children		2.6 (1.74)
Monthly income (RM)		4213.7 (1891.95)
Employment (year)		15.7 (7.58)
Yearly performance mark		
Satisfied	284 (90.4)	
Dissatisfied	30 (9.6)	

**Table 3 ijerph-19-16106-t003:** Proportion of job dissatisfaction among healthcare workers working in different health clinics’ working systems (*n* = 314).

Variable	Proportion	
	Job Satisfaction (%)	Job Dissatisfaction (%)
Health clinic’s working system		
Shift	103 (32.8)	69 (22.0)
Non-shift	99 (31.5)	43 (13.7)

**Table 4 ijerph-19-16106-t004:** Proportion of various levels of satisfaction towards job-related factors among healthcare workers in ‘Type 2 Health Clinics’ (*n* = 314).

Job-Related Factors	No. of Dissatisfied (%)	No. of Satisfied (%)
Pay	94 (29.9)	220 (70.1)
Promotion	148 (47.1)	166 (52.9)
Supervision	53 (16.9)	261 (83.1)
Benefits	174 (55.4)	140 (44.6)
Rewards	134 (42.7)	180 (57.3)
Operating conditions	194 (61.8)	120 (38.2)
Co-workers	48 (15.3)	266 (84.7)
Nature of work	27 (8.6)	287 (91.4)
Communication	110 (35.0)	204 (65.0)

**Table 5 ijerph-19-16106-t005:** Simple and multiple logistic regression analysis for factors associated with job dissatisfaction among healthcare workers working in ‘Type 2 Health Clinics’ in north-eastern Malaysia.

Variables	Crude OR ^a^ (95% CI)	*p*-Value ^a^	Adjusted OR ^b^ (95% CI)	*p*-Value ^b^
Age (year)	0.953 (0.882, 1.029)	0.214	0.906 (0.826, 0.994)	0.037
Gender				
Male	1			
Female	4.603 (0.585, 36.226)	0.147		
Race				
Malay	1			
Non-Malay	5.528 (0.534, 57.211)	0.152		
Marital status				
Single/divorced	1			
Married	1.392 (0.172, 11.250)	0.756		
Education level				
Tertiary	1			
Secondary	0.842 (0.249, 2.845)	0.782		
Health clinic’s working system				
Non-shift	1			
Shift	0.824 (0.268, 2.537)	0.736		
Number of children	0.860 (0.612, 1.208)	0.384		
Monthly income (RM)	1.000 (0.999, 1.000)	0.139		
Employment (year)	0.963 (0.891, 1.040)	0.337		
Yearly performance mark				
Satisfied	1		1	
Dissatisfied	7.792 (2.266, 26.789)	0.001	14.795 (3.433, 63.759)	<0.001

^a^ Simple Logistic Regression; ^b^ Multiple Logistic Regression; Constant = 0.601; Forward LR and Backward LR method; No interaction and no multicollinearity; Hosmer–Lemeshow Test is not significant, *p*-value = 0.465; Classification Table 93.5%; Area Receiver Operating Characteristics (ROC) 76.4%.

## Data Availability

There is no reported data.

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
