# Peer review of "Job Dissatisfaction and Its Predictors among Healthcare Workers of ‘Type 2 Health Clinics’ in North-Eastern Malaysia"

_ijerph, 2022, doi:10.3390/ijerph192316106_

Round 1

Reviewer 1 Report

Job Dissatisfaction and Its Predictors among Healthcare Work-  ers of Type 2 Health Clinic in North-Eastern Malaysia

Abstract

First.The summary should be more synthetic and structured in a way that includes the following clearly:

The contextualization of the study

(The main objective)

(The justification)

The sample used

The methods used

The main findings and conclusions

(The novel contribution)

What has been written in red above between brackets  should be briefly added. 

Second. There are a lot of results and numbers. It is better to move them to the results section

Introduction

1.Introduction segments are not well connected. 

 2.The introduction of the paper doesn’t describe the problem within a theoretical framework. It's not clearly established,  doesn’t identify a gap, and doesn’t occupy the gap.

3. The introduction needs to be more clear and straight to point by justifying soundly why your study includes Job Dissatisfaction and Its Predictors among Healthcare Workers.

 4.  What is the importance of this topic for North-Eastern Malaysia? How did you go beyond the past work?

5. The authors also need to justify the unique contributions of this paper in more depth.

Literature Review

The Literature section needs to be added as a separate headline . While establishing the hypotheses or the questions, the authors must give an extensive background; it needs a comprehensive review to justify the proposed study.  

The sources cited are not enough. The research gaps in the previous studies related to this topic were not explained. Please address these issues.

Methodology.

1.      Procedures have not  been described appropriately.

2.      How did the authors contact the respondents. Explain how?

3.      What is the response rate? Does the authors' sample represent the population? Make sure whether the authors' sample can represent the population. The authors need to justify how they tested the validity and reliability clearly at the methodology section.

Results

fine

Discussions

1.The discussions are somewhat broad and general. I can’t see the clear explanation for this argument. It is complex to follow the line of arguing and to identify how each of the analyses and their results justifies the discussion.

2. There are mixed between Discussions and conclusions like the limitations and the future study.

The conclusions

Need to include a clear theoretical implication and practical implication of the research. Also limitations and the future studies.

Others:

The paper has some editing issues. It needs proofreading.

Good luck

Reviewer 2 Report

Reviewer Comments

Full Title: Job Dissatisfaction and Its Predictors among Healthcare Workers of Type 2 Health Clinic in North-Eastern Malaysia

Manuscript Number: ijerph-1999314

Dear author,

Thank you for your work. Your work focuses on the factors that affect job dissatisfaction among healthcare workers. A survey was conducted to analyze these factors and their effect.

Below, please find my comments which I believe should be addressed before this work can be published.

Title:

·         Clinic: I believe it should be Clinics.

Abstract

·         The abstract needs rewriting and structuring. I believe it needs improvement to answer the following points (those indicated with NOT Okay):

o   What is the problem? okay

o   Why is it a problem? Okay

o   Targeted population? In other words, what do you mean by Type 2 workers? NOT Okay

o   What is the purpose of your work? Okay

o   The methodology you followed to get these lessons. Okay

o   key results and conclusions. Not Okay. The key results (Lines 15-19) need further explanation. For example, what do you mean by 35.7%?

·         English is good

·         Keywords! (I suggest replacing “factor associated” with “Type 2 workers”). It is up to you.

·         The abstract is an essential slice of the article; it decides if the reader will continue reading your paper or not. I recommend you improve it more.

1. Introduction

·         English proofreading should be conducted. There is a lot to improve, and here are a few examples just from the first paragraph

o   “According to Family Health Development … Malaysia under …” is missing a comma before under

o   Ln 26: “…health clinic can be divided ..” should be “ … health clinics can be divided …”

o   Ln 27: “types depend on its …” should be “types depending on its …”

o   Ln 27: “…. Per day.” Per day should be deleted.

·         At the end of the introduction, I was expecting to see this work's punchy aims and contributions! State the contribution and value of this work.

·         Please, define type 2 workers. You also mentioned type 2 health clinic! Is it the same?

2. Literature Review

·         The literature review needs improvement. It could be further extended, there are too many related studies. I found a few studies that could be related or useful to mention (if you see fit, I looked for recent studies quickly)

o   Sustainability | Free Full-Text | Decision-Making Framework for Evaluating Physicians’ Preference Items Using Multi-Objective Decision Analysis Principles (mdpi.com)

o   Healthcare | Free Full-Text | Development of an Evidence-Informed Solution to Emotional Distress in Public Safety Personnel and Healthcare Workers: The Social Support, Tracking Distress, Education, and Discussion CommunitY (STEADY) Program (mdpi.com)

o   IJERPH | Free Full-Text | Factors Affecting the Career Continuation of Newly Graduated and Reinstated Dental Hygienists Who Participated in a Technical Training Program in Japan (mdpi.com)

o   IJERPH | Free Full-Text | Job Crafting in Nursing: Mediation between Work Engagement and Job Performance in a Multisample Study (mdpi.com)

3. Results

·         What are your hypotheses?

·         The Data Analysis subsection is worth mentioning the statistical tools and techniques you utilized and why?

·         A graphical representation of some key results could be shown.

4. Discussion and Conclusion

·         Study limitations were mentioned. Okay

·         Future work could be moved to the conclusion section (suggestion).

Reviewer 3 Report

The abstract must be completely reviewed. It´s not clear the scientific gap, the novelty, and the practical and scientific contribution of the paper.

Please, in the Introduction frame the questions being addressed, describe the methods briefly, and provide context for the findings being presented. Present a clear research question and convince the reader that your work fills a gap in current knowledge.

The state of the art and the practical and scientific contribution of the paper is vague.

The reference Manan et al (2015 or 2012?) is a self-citation

The section Materials and Methods doesn't describe the analysis methods. Why simple and multiple logistic regression analysis were used. What are the differences between the results?

What is a type 2 health clinic?

Line 81: It has nine facets and a 36-item scale to evaluate employee attitudes toward the job and its various aspects. The facets were pay, promotion, contingent rewards, operating procedures, supervision, nature of work, fringe benefits, communication, and co-workers.

Where are the 9 facets results? Why they were not discussed?

Insert the questionnaire as an Appendix.

Round 2

Reviewer 1 Report

I agree with all  the comments , but  the abstract  and  point 3 still need to be improved

The abstract is  fine now but  still  long, . I suggest that it be  shortened  

point 3: Literature Review

-The Literature section needs to be added as a separate headline . While establishing the hypotheses or the questions, the authors must give an extensive background; it needs a comprehensive review to justify the proposed study

 I believe that the authors mixed between the introductions and the Literature review ...  

you can make it more clear and simple 

Author Response

  1. To address comments regarding the abstract, it had been reviewed and simplified.
  2. The new section of section 2. Literature Review’ had been added to discuss further the background of this topic

Reviewer 3 Report

Although the authors modified substantially the main text, some aspects of the paper still must be improved.

Line 119: It has nine facets and a 36-item scale to evaluate employee attitudes toward the job and its various aspects. The facets were pay, promotion, contingent rewards, operating procedures, supervision, nature of work, fringe benefits, communication, and co-workers.

Where are the 9 facets results? Why the authors discussed only sociodemographic aspects?

Why the other facets were not discussed?

Author Response

We are providing a new table of proportion of facets in JSS and will be discuss it in the discussion section

Round 3

Reviewer 3 Report

Although the authors considered the nine facets analysis and discussion about these facets must be improved. The questionnaire has 2 parts: Demographic and Job Satisfaction, but the study is very limited to the first part of the questionnaire.

Line 184:

The authors only presented a table with the results limited to % satisfied or % dissatisfied, not even performing a basic statistical analysis evaluating the differences between the results.

Please, please link the demographic evaluations with the 9 facets through appropriate statistical analysis.

Line 266:

This analysis must be improved and must also be deepened and supported by appropriate statistical analysis.

Appendix: Please, put S1 – S12 in English.

Author Response

Thank you for your input.

After extensive analysis and discussion, we are deciding to not include the analysis of each facet (domain of the questionnaire) with sociodemographic factors as suggested because:

1) The analysis will be very extensive.

2) The results will also be extensive and change the discussion part and conclusion. As the initial purpose of this manuscript is to address the factors associated with job dissatisfaction not to analyze each domain of job dissatisfaction that contributed to job dissatisfaction.

3) We hope the current content in the manuscript will help the readers understand our research.